# ADVANCING PORTFOLIO OPTIMIZATION: HYBRID RELAXATION AND HEURISTIC APPROACHES FOR CARDINALITY CONSTRAINED MIQP PROBLEMS

## ABSTRACT

The growing magnitude of investments in global markets has intensified the need for sophisticated risk mitigation strategies in portfolio optimization. Traditional portfolio optimization models that seek to minimize risk for a specified return frequently incorporate cardinality constraints, rendering them as Mixed-Integer Quadratic Programming (MIQP) challenges. These constraints elevate the problem to NP-Hard status, complicating the solution process. While heuristic methods have historically been favored for their direct approach to MIQP problems, relaxation techniques offer a strategic alternative by simplifying MIQP into a more tractable Quadratic Programming (QP) problem. We first introduce an approach that facilitates the conversion of MIQP to QP by relaxing integer constraints into continuous domains and integrating integer conditions into the objective function using Lagrange multipliers. This dual application not only eases the computational burden but preserves the integrity of the original problem's structure. An innovative diagonalization technique applied to the covariance matrix further refines our method, enhancing the fit for integer variables, as Lagrange multipliers are inherently biased towards continuous variables. We present a comparative analysis of three distinct models, Linear, Dual, and Diagonal or Diag, each employing a unique relaxation strategy. Our research evaluates their efficacy in addressing the MIQP problem under cardinality constraints. In conjunction with heuristic methods, the refined solutions from our exact relaxation models serve as a starting point for further refinement using Genetic Algorithm and Neighborhood Searching Algorithm. This hybrid methodology yields results that not only rival but occasionally surpass those achieved by the latest models and the commercial solver CPLEX. Our findings endorse the potential of combining exact and heuristic techniques in portfolio optimization, marking a significant advancement in the field.

## 1    INTRODUCTION

With trillions of dollars circulating in global stock markets, investors are increasingly seeking portfolios that not only generate strong returns but also mitigate risk, making their investments safer. The challenge lies in identifying the optimal combination of assets that maximizes expected returns while minimizing risk, often measured by the variance of the portfolio. Striking the right balance between return and risk is crucial for developing effective portfolio strategies.

Markowitz (1952) introduced a model that examines the relationship between expected return and risk, measured by variance, based on the different weights assigned to securities or assets. This mean-variance (MV) model uses historical prices to estimate expected returns and asset correlations to calculate variance. By optimizing either the sum of expected returns or minimizing total variance, the optimal weight for each asset can be determined. Later, Markowitz (1956) extended this work by providing an optimization approach for quadratic functions, specifically variance, subject to linear constraints and a fixed expected return. This formulation, a Quadratic Programming (QP) problem, also introduced the concept of "efficient points," which form what is now known as the Efficient Frontier (EF).

While the Mean-Variance (MV) model is foundational, it lacks realism without incorporating practical constraints. For instance, Grazia Speranza (1996) introduced minimum transaction units and bounds on asset weights into the model, and Chang et al. (2000) emphasized the importance of restricting the number of selected assets, making the model more applicable to real-world scenarios. The inclusion of constraints, particularly cardinality constraints, transforms the MV model from a Quadratic Programming (QP) problem into a Mixed-Integer Quadratic Programming (MIQP) problem, which is NP-Hard (Pia et al., 2017). Several studies, including Guzelsoy & Ralphs (2007), Guzelsoy & Ralphs (2011), and Feizollahi et al. (2017), have conducted extensive research on MIQP using Lagrange duality models. Building on this work, Shaw et al. (2008) explored how embedding the constraints into the original primal model through Lagrange multipliers forms a relaxed duality model or Dual Model. Although these relaxations introduce errors and gaps in the solutions, they make it feasible to find solutions within a practical time frame. Shaw et al. (2008) demonstrated a significant reduction in solution time using CPLEX, and further research by Xu et al. (2024) showed that the Dual Model can help establish a lower bound for the primal, facilitating solution approaches. In addition to Dual Models, various heuristic methods have been proposed to tackle more realistic models with constraints.

Numerous heuristic and meta-heuristic approaches have been applied to MIQP portfolio optimization. Woodside-Oriakhi et al. (2011) explored Genetic Algorithms, Tabu Search, and Simulated Annealing metaheuristics, achieving better results than previous heuristic methods. Building on this, Deng et al. (2012) introduced an Improved Particle Swarm Optimization (PSO) technique, which enhanced the robustness and effectiveness of portfolios, particularly in low-variance conditions. Similarly, Lwin & Qu (2013) achieved competitive results using a Hybrid Algorithm. Further advancements came with the Artificial Bee Colony (ABC) Algorithm, where Tuba & Bacanin (2014) demonstrated a smaller Euclidean distance between solutions compared to prior methods. This approach was later refined by Kalayci et al. (2017), who added a feasibility enforcement procedure to the ABC Algorithm. Recently, Kalayci et al. (2020) reintroduced the Hybrid Algorithm, constructing an efficient metaheuristic combining Ant Colony Optimization, Genetic Algorithms, and ABC Optimization, yielding promising results. In addition to the growth of ABC-based methods, Baykasoğlu et al. (2015) developed a Greedy Randomized Adaptive Search Procedure (GRASP), while Ertenlice & Kalayci (2018) explored swarm intelligence (SI) to address time complexity issues. Furthermore, Akbay et al. (2020) proposed a new method, the Parallel Variable Neighborhood Search Algorithm, which demonstrated high efficiency in solving MIQP problems.

Although these heuristic methods have gained relatively high results, most only focus on the original primal problem, and their relaxation models also only respect the Primal Model. However, for Primal Model itself, it is not a solvable model as NP-Hard, and those heuristic methods also cannot guarantee for the solutions. Here is where exact methods are introduced. That is, the cardinality constraint could be relaxed into a linear constraint, which transforms the MIQP into a solvable QP. Once the model is solved, the modified linear constraint then could be discretized back into the original cardinality constraint. That would be a logical way for solving MIQP models. Considering all the constraints including the cardinality constraint, using Lagrange multipliers, all constraints are able to be embedded into the objective function, transforming it into a Dual Model. By pre-solving the cardinality constraint or deriving its optimal values ahead to further refining the process, this allows us MIQP to be solved without the cardinality constraint directly. The Lagrange approach helps derive the expressions for the integer variables. Previous work by Fisher (1981); Nemhauser & Wolsey (1988) has demonstrated that this Dual Model effectively relaxes the problem, offering a pathway to approach the solution by solving with respecting to the multiplier variables. Additionally, Li et al. (2006); Shaw et al. (2008) have provided substantial evidence supporting the efficiency of this method. Furthermore, recognizing that Lagrange multipliers are more suited for continuous variables, potential inefficiencies might happen when dealing with integer variables. To mitigate errors introduced by the multipliers for discrete variables, a employment of a more advanced relaxation could help the model approach the solution better. Xu et al. (2024) proposed a method using the diagonalization of the covariance matrix, establishing lower bounds for the diagonal matrix in correlation and ensuring feasible solutions for MIQP.

Analyzing MIQP is not only valuable for portfolio optimization but also has broad applications across various fields. For example, the most recent advancements include a real-time, mixed-integer programming-based decision-making system for automated driving (Quirynen et al., 2024), and optimization in scheduling, such as solving job-shop scheduling problems (Ajagekar et al., 2022).

MIQP is also increasingly relevant in machine learning, where it is used for solving decision problems in which the objective function is a machine learning model (Anderson et al., 2020), processing images and conducting gap analysis (Wang, 2022; 2024), and embedding the model as a layer within neural networks to improve training outcomes (Ferber et al., 2020).

In this paper, we introduce a trio of exact relaxation techniques for the cardinality constrained Mixed-Integer Quadratic Programming (MIQP) problem: the Linear Model, Dual Model, and Diag Model, corresponding to linear, dual, and diagonal relaxations of the primal problem, respectively. These methodologies offer a novel perspective on addressing the complexities inherent in MIQP. To augment the precision of these exact relaxations, we have innovatively combined them with heuristic methods, thereby crafting an approach that leverages the strengths of both exact and heuristic optimization techniques. Our comprehensive evaluation extends beyond mere application; we meticulously define and examine the optimality gaps between the solutions of the primal problem and those of the relaxed models. This gap analysis is critical for understanding the effectiveness of our proposed models and for quantifying the enhancements provided by the heuristic methods. The synthesis of exact relaxations with heuristic strategies, as presented in this work, showcases a significant advancement in the pursuit of solving MIQP problems. The integration not only fortifies the theoretical underpinnings of the relaxation models but also amplifies their practical application. Through rigorous testing and gap analysis, we demonstrate that our approach yields a more robust and effective means of navigating the complex landscape of MIQP, setting a new benchmark for future research in the field.

The remainder of this paper is organized as follows: In Section 2, we formally define and formulate our models and gaps for later justification, and in Section 3 we state the detailed approach to how we get our results using the models. Then in Section 4 we run our approach and examine the effectiveness of our method. For Section 5 we drop out the conclusion with our results.

## 2 PROBLEM FORMULATION

### 2.1 PRIMAL MODEL

For Primal Model, or original cardinality constrained portfolio optimization problem, it can be defined as:

$$\min_{w_i} v = \sum_{i=1}^{n} \sum_{j=1}^{n} C_{i,j} w_i w_j,$$

$$\text{s.t.} \sum_{i=1}^{n} r_i w_i = r,$$

$$\sum_{i=1}^{n} w_i = 1, \tag{1}$$

$$b_i \in \mathbb{B},$$

$$\sum_{i=1}^{n} b_i = k,$$

$$l_i b_i \leq w_i \leq u_i b_i, \qquad i = 1, \ldots, n.$$

where $n$ is the total number of assets; $C_{i,j}$ is the element $i, j$ of the covariance matrix $\boldsymbol{C}$, $w_i$ is the element i of the weight vector $\boldsymbol{w}$, the weight of $i$-th element in the list, $\sum_{i=1}^{n} \sum_{j=1}^{n} C_{i,j} w_i w_j$ is the sum of the variance or risk $v$ and $\min_{w_i}$ shows that we want to minimize it respecting to $\boldsymbol{w}$; $r_i$ is the element $i$ of the expected return vector $\boldsymbol{r}$, $\sum_{i=1}^{n} r_i w_i = r$ means the sum of weighted return should be the expected total return $r$; $\sum_{i=1}^{n} w_i = 1$ means the sum of weight for all elements should be 1; for $b_i \in \mathbb{B}$ we define a binary variable $b_i$ in the binary domain $\mathbb{B}$ where the asset $i$ is chosen if $b_i = 1$, otherwise $b_i = 0$; $\sum_{i=1}^{n} b_i = k$ refer to the total number of assets we would choose is $k$; and $l_i b_i \leq w_i \leq u_i b_i$ we bound the weight for every asset with $b_i$, meaning if $b_i = 0$ we force the weight to 0 since the asset is not chosen.

## 2.2 LINEAR MODEL

For Linear Model, or linear relaxation on the Primal Model, the cardinality constraint has been relaxed from integer or binary numbers to real numbers. This could turn the MIQP to QP making the problem solvable. That is, the integer or binary variables $b_i$ is linearized to $b_i^l$. After solving $b_i^l$ on the solvable model, $b_i^l$ is discretized back to $b_i$ getting the relaxed solution:

$$
\begin{aligned}
\min_{w_i} v = & \sum_{i=1}^n \sum_{j=1}^n C_{i,j} w_i w_j, \\
\text{s.t.} & \sum_{i=1}^n r_i w_i = r, \\
& \sum_{i=1}^n w_i = 1, \\
& b_i^l \in [0,1], \\
& \sum_{i=1}^n b_i^l = k, \\
& l_i b_i^l \le w_i \le u_i b_i^l, \qquad i = 1, \ldots, n, \\
& b_i = \mathbf{1}_{b_i^l \in \text{top}^k \{b^l\}}.
\end{aligned}
\tag{2}
$$

where the formulas are defined similarly to the Primal Model with a replacement of $b_i$ by linear temporary variables $b_i^l$ ranged from 0 to 1; and $b_i = \mathbf{1}_{b_i^l \in \text{top}^k \{b^l\}}$ implies that $b_i^l$ is tuned with the top $k$ elements to 1 and the rest are 0 to form $b_i$. In general, we linearize $b_i$ to $[0,1]$ variables, then discretize them back to $\mathbb{B}$ after choosing the top $k$ results.

## 2.3 DUAL MODEL

For Dual Model, we introduce Lagrange multipliers:

$$
\max_{\alpha_i, \beta_i \ge 0, \lambda_0, \lambda_1, \lambda_2 \in \mathbb{R}} \min_{w_i \in \mathbb{R}, b_i \in \mathbb{B}} \sum_{i=1}^n \sum_{j=1}^n C_{i,j} w_i w_j + \lambda_1 (\sum_{i=1}^n r_i w_i - r) + \lambda_2 (\sum_{i=1}^n w_i - 1)
$$
$$
+ \lambda_0 (\sum_{i=1}^n b_i - k) + \sum_{i=1}^n \alpha_i (w_i - u_i b_i) + \sum_{i=1}^n \beta_i (l_i b_i - w_i)
\tag{3}
$$

where we introduce scalar multipliers $\lambda_1$, $\lambda_2$, $\lambda_0$, and vector multipliers $\boldsymbol{\alpha}$, $\boldsymbol{\beta}$, after minimizing $w_i \in \mathbb{R}, b_i \in \mathbb{B}$, we can derive $w_i = \frac{1}{2} \sum_{j=1}^n C_{i,j}^{-1} (\beta_j - \alpha_j - \lambda_1 r_j - \lambda_2)$, then get:

$$
\max_{\alpha_i, \beta_i \ge 0, \lambda_0, \lambda_1, \lambda_2 \in \mathbb{R}} - \frac{1}{4} \sum_{i=1}^n \sum_{j=1}^n (\beta_i - \alpha_i - \lambda_1 r_i - \lambda_2) C_{i,j}^{-1} (\beta_j - \alpha_j - \lambda_1 r_j - \lambda_2)
$$
$$
+ \sum_{i=1}^n (\beta_i l_i - \alpha_i u_i + \lambda_0) - \lambda_0 k - \lambda_1 r - \lambda_2
\tag{4}
$$
$$
\text{s.t.} \beta_i l_i - \alpha_i u_i + \lambda_0 \le 0.
$$
$$
b_i = \begin{cases} 1, & \beta_i l_i - \alpha_i u_i + \lambda_0 < 0 \\ 0, & \beta_i l_i - \alpha_i u_i + \lambda_0 = 0 \end{cases}
$$

## 2.4 DIAG MODEL

In order to cover all the primal solutions using gaps, some relaxations is preferred for their wider range with a small increase of time finding the solutions. Thus, we introduce a new dummy covariance matrix $D$ with same apposition where $w^T(D - C)w \leq 0$ for any $w \in \mathbb{R}^n$:

$$D_{i,j} = \begin{cases} 0, & i \neq j \\ \frac{1}{\sum_{k=1}^n |C_{i,k}^{-1}|}, & i = j \end{cases} \tag{5}$$

Since we need the gap not to be too large, we just want to partially substitute this dummy matrix, so we insert it into our previous Dual Model:

$$\max_{\alpha_i, \beta_i \geq 0, \lambda_0, \lambda_1, \lambda_2 \in \mathbb{R}} - \sum_{i=1}^n \sum_{j=1}^n D_{i,j} w_i w_j + \sum_{i=1}^n (\beta_i l_i - \alpha_i u_i + \lambda_0) - \lambda_0 k - \lambda_1 r - \lambda_2$$

$$\text{s.t.} w_i = \frac{1}{2} \sum_{j=1}^n C_{i,j}^{-1} (\beta_j - \alpha_j - \lambda_1 r_j - \lambda_2) \tag{6}$$

$$\beta_i l_i - \alpha_i u_i + \lambda_0 \leq 0.$$

$$b_i = \begin{cases} 1, & \beta_i l_i - \alpha_i u_i + \lambda_0 < 0 \\ 0, & \beta_i l_i - \alpha_i u_i + \lambda_0 = 0 \end{cases}$$

With a deeper analysis, we are also able to bound the Dual Model with a lower bound:

$$\max_{\alpha_i, \beta_i \geq 0, \lambda_0, \lambda_1, \lambda_2 \in \mathbb{R}} - \frac{1}{4} \sum_{i=1}^n \sum_{j=1}^n (\beta_i - \alpha_i - \lambda_1 r_i - \lambda_2) D_{i,j}^{-1} (\beta_j - \alpha_j - \lambda_1 r_j - \lambda_2)$$

$$+ \sum_{i=1}^n (\beta_i l_i - \alpha_i u_i + \lambda_0) - \lambda_0 k - \lambda_1 r - \lambda_2 \tag{7}$$

$$\text{s.t.} \beta_i l_i - \alpha_i u_i + \lambda_0 \leq 0.$$

$$b_i = \begin{cases} 1, & \beta_i l_i - \alpha_i u_i + \lambda_0 < 0 \\ 0, & \beta_i l_i - \alpha_i u_i + \lambda_0 = 0 \end{cases}$$

which is almost the same as the Dual Model but with the total substitution of $C$ with $D$. With a further relaxation, the Lagrange multipliers, which used to fit with continuous variables the best, are able to cover the integer or binary variables and approach the solution better.

## 3 PROPOSED SOLUTION APPROACH

To solve the problem, we employ a multistep approach integrating several methods. First, the Primal Model is relaxed into QP models. Starting with random initial solutions, with solutions generated by the QP models, an environment is created. Second, a Genetic Algorithm (GA) is applied to this environment, where a series of fitness evaluations are conducted, and the best GA solution is selected for further refinement. Third, the solution is improved using a neighborhood search algorithm, which enhances its accuracy. Finally, the refined solution from the neighborhood search is used to restrict the original MIQP to a QP formulation, yielding the optimal asset weights. The following sections provide a detailed explanation of each step.

### 3.1 ORIGINAL SOLUTIONS GENERATION

For the input parameters, we require the expected returns for each asset and the covariance matrix between assets. Additionally, we must specify the target total return, the number of assets to select, and the upper and lower bounds for asset weights.

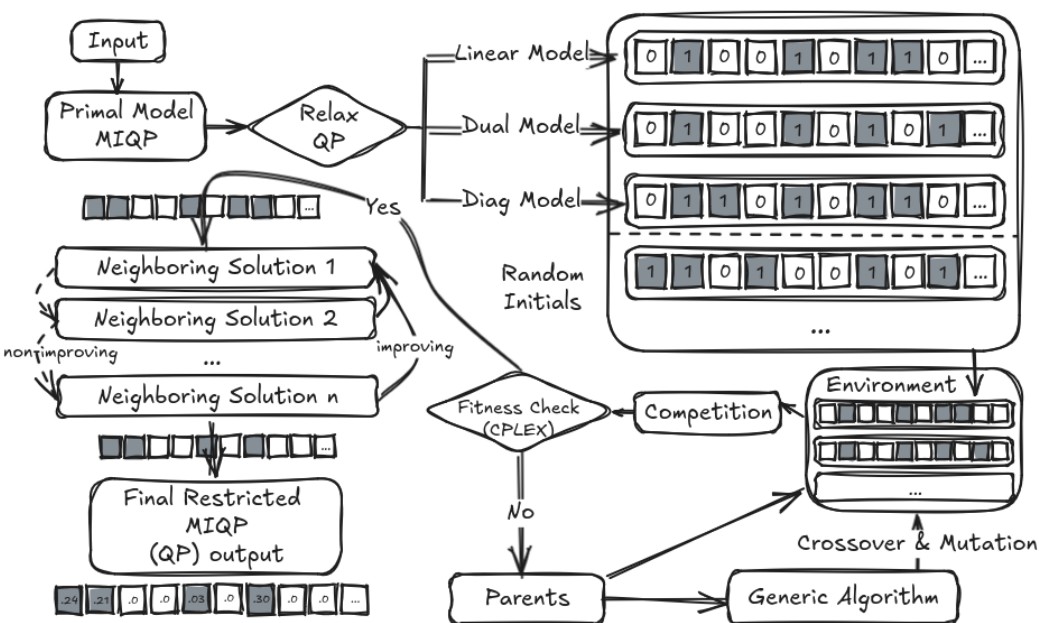

Figure 1: A schematic visualizing our approach. Supported by Open-Source Excalidraw.

Given that CPLEX cannot efficiently handle large-scale MIQP problems, we first relax the Primal MIQP Model into three models: the Linear Model, Dual Model, and Diag Model. Using the predefined parameters, we can generate initial solutions from these relaxed models. Simultaneously, we generate $M^{\text{Random}}$ random initial solutions following these rules: the solution vector size should be $n$, each element in the vector must be either 0 or 1, exactly $k$ values in the vector should be 1, and each value has an equal probability of being 1. This gives us a total of $M = M^{\text{Relax}} + M^{\text{Random}}$ solutions, which are then assigned to the environment for further processing.

## 3.2 GENETIC ALGORITHM APPROACH

After generating solutions within the environment, we evaluate their competitiveness using a fitness function. This fitness function is defined as follows: given a solution vector $\boldsymbol{b}$, we introduce a new constraint into the Primal Model, requiring that the selected variables in the Primal Model match the values in $\boldsymbol{b}$. This transforms the problem into a QP. By solving this QP, we obtain the variance $v$, which is used as the measure of competitiveness. The fitness function is therefore $f(\boldsymbol{b}) = v$, where a lower value of $f(\boldsymbol{b})$ indicates a more competitive solution.

Using this fitness function, we conduct a competition among the solutions in the environment. We retain the top $p\%$ of the solutions based on fitness and discard the rest. Next, we perform a fitness check, where the difference between the worst and best solutions must not be too large. Specifically, we require that $l \geq (\max\{f(\boldsymbol{b})|\boldsymbol{b} \in \text{environment}\} - \min\{f(\boldsymbol{b})|\boldsymbol{b} \in \text{environment}\})/\min\{f(\boldsymbol{b})|\boldsymbol{b} \in \text{environment}\}$ for solutions within the environment. If this condition is met, the best solution is selected for the next step. Otherwise, two-parent solutions are randomly selected, and we apply a crossover operation to generate a new solution.

The crossover is performed under the following rules: the new solution vector size is $n$, each element in the vector must be either 0 or 1, and exactly $k$ elements should be 1. If both parents have $b_i^{\text{father}} = b_i^{\text{mother}} = 1$, then $b_i^{\text{son}} = 1$. For positions where $b_i^{\text{father}} \neq b_i^{\text{mother}}$, $b_i^{\text{son}}$ is assigned a value of 1 with equal probability. The new solution is also subjected to a mutation with a probability of $m\%$, where one index with $b_i = 1$ and another with $b_i = 0$ are swapped.

Once the new solution is generated, it, along with its parent solutions, is reintroduced into the environment, and the competition is repeated. This process continues iteratively until the fitness check condition is satisfied.

### 3.3 Neighborhood Search Algorithm Approach

With the best solution generated by the Genetic Algorithm (GA) approach, which is either very close to or equal to the optimal primal solution, we can apply neighborhood searching to refine the solution further. In this approach, the GA solution is initially treated as the "neighborhood solution" or neighboring solution 1.

To generate neighboring solution 2, we randomly select two values in the neighborhood solution: one with a value of 1 and another with a value of 0. We then swap their positions. After that, we compare the fitness of neighboring solution 2 with that of the initial neighborhood solution. If the fitness of neighboring solution 2 is better (i.e., lower), it becomes the new neighborhood solution, and we repeat the process.

If neighboring solution 2 is not better, we proceed by generating neighboring solution 3 in the same manner—swapping the values of 1 and 0 based on neighboring solution 2 and comparing it with the neighborhood solution. This iterative process continues until either an improvement is found or a predefined limit is reached.

## 4 Experiment Results

### 4.1 Dataset Description

Our approach and later gap analysis are based on weekly price data from March 1992 to September 1997 of the Hang Seng (Hong Kong), DAX 100 (Germany), FTSE 100 (UK), S&P 100 (USA) and Nikkei 225 (Japan) (Chang et al., 2000) from OR library. The size of each dataset is ranged from $n = 31$ for Hang Seng to $n = 225$ for Nikkei. We set the global upper bound for each asset as $u_i = 1$ and the lower bound as $l_i = 0.01$, and the total number of assets we would choose is $k = 10$.

### 4.2 Efficient Frontiers and Percentage Errors

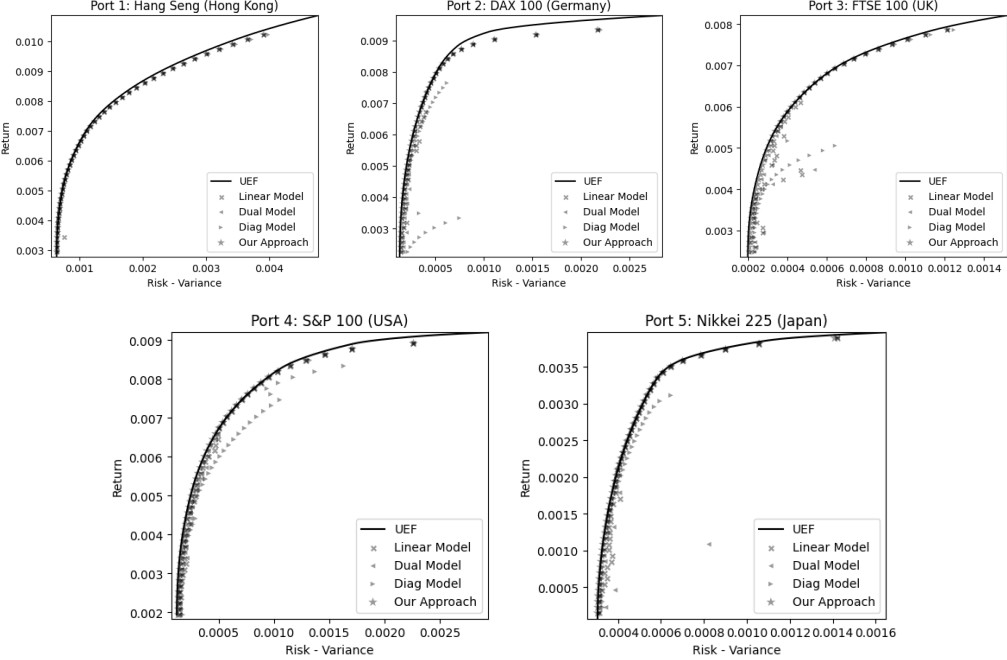

Figure 2: EF and solutions generated by different approaches. Supported by Matplotlib.

Based on 5 datasets above, the given unconstrained efficient frontier (UEF) as an upper bound is drawn as a line in each figure. For comparing and visualizing the effect of Linear, Dual, Diag, and

Ours Models, we sample 50 points respecting to returns from the domain of the UEF and then mark the points on each figure. In all the figures, the risks have a positive correlation with returns, which means higher returns come with higher risks, and our approach based on Linear, Dual, and Diag Models should remain left which is the closest to the upper bound line. The features shown in the figures logically make sense.

Table 1: Percentage Errors of different approaches

| PE | | CPLEX | GA | TS | SA | Linear | Dual | Diag | Ours |
|---|---|---|---|---|---|---|---|---|---|
| Port 1 | Mean | **0.5208** | 0.8510 | 0.8234 | 1.0589 | 0.7929 | 0.5232 | 0.9131 | 0.5231 |
| | Median | 0.4147 | 0.5873 | 0.3949 | 0.5355 | 0.4317 | 0.4147 | 0.6223 | **0.3356** |
| Port 2 | Mean | 1.6584 | 0.7740 | 0.7190 | 1.0267 | 4.8232 | 5.8435 | 10.3267 | **0.6050** |
| | Median | 1.2513 | 0.2400 | 0.4298 | 0.8682 | 4.3321 | 2.9218 | 6.0821 | **0.1502** |
| Port 3 | Mean | 1.2011 | 0.1620 | 0.3930 | 0.8952 | 4.7993 | 4.0280 | 6.2230 | **0.1598** |
| | Median | 2.8221 | **0.0820** | 0.2061 | 0.3944 | 5.3894 | 2.8866 | 9.7862 | 0.3217 |
| Port 4 | Mean | 2.7642 | 0.2922 | 1.0358 | 3.0952 | 5.5428 | 5.8134 | 8.9946 | **0.2272** |
| | Median | 2.6479 | **0.1809** | 1.0248 | 2.1064 | 4.2753 | 5.3949 | 11.0371 | 0.9978 |
| Port 5 | Mean | 0.1417 | 0.3353 | 0.7838 | 1.1193 | 1.3079 | 1.7143 | 2.0529 | **0.1368** |
| | Median | **0.1039** | 0.3040 | 0.6525 | 0.6877 | 0.4247 | 0.3827 | 1.7317 | 0.1363 |

Then Table 1 reflects the percentage error from point to the EF line in the figures. In the table, Linear, Dual, and Diag models as relaxation models cannot always have a good effect on solution generation. But our approach based on these exact solutions, and with heuristics, gains a good effect. At the same time, we introduce more heuristics like Genetic Algorithm (GA), Tabu search(SA), and Simulated Annealing (SA) to compare and verify our approach. Compared with CPLEX optimal solutions and others heuristics (Woodside-Oriakhi et al., 2011), our approach gains a close but mostly a better effect. To be more specific, mostly we reach a mean percentage errors, implying that our method has fewer outliners and thus has a more stable results.

To be more specific, our approach outperforms the solutions in larger datasets. For portfolio 2, we achieve a value of 0.6050, for portfolio 3, 0.1598, for portfolio 4, 0.2272, and for portfolio 5, 0.1368, in terms of the mean. With respect to the median, we attain the best result for portfolio 2 with 0.1502 and maintain strong performance across other datasets, even though we do not always achieve the best result.

### 4.3 MODELS GAP FORMULATION

Since we always compare and analyze our gaps between one of the relaxed models and CPLEX optimal Model, we define the gaps for them as $g^{\text{Relaxed}} = b^{\text{Relaxed}} - b^{\text{CPLEX}}$, where $b^{\text{Relaxed}}$ stands for the choice binary variables' solution for one of the relaxed models, and $b^{\text{CPLEX}}$ stands for CPLEX optimal's. Since $d^{\text{Relaxed}}$ is a differences vector between two binary vectors, we define variable $g^{\text{Relaxed}} = \frac{1}{2}|g^{\text{Relaxed}}| = \frac{1}{2}\sum_{i=1}^{n}|b_i^{\text{Relaxed}} - b_i^{\text{CPLEX}}|$ to analyze the number of all differences between two solutions. To be more specific:

$$g^{\text{Linear}} = b^{\text{Linear}} - b^{\text{CPLEX}}$$
$$g^{\text{Dual}} = b^{\text{Dual}} - b^{\text{CPLEX}} \tag{8}$$
$$g^{\text{Diag}} = b^{\text{Diag}} - b^{\text{CPLEX}}$$

In this way, we are able to quantify and analyze the binary gaps. Our results show in Table 2.

The Table 2 reflects the relations between solution generated by relaxed models and our models with CPLEX Optimal solution. Most of them reflects that our approaches is closer to CPLEX optimal

Table 2: Gaps of different approaches

| Gaps | | Linear | Dual | Diag | Ours |
|---|---|---|---|---|---|
| Port 1 | Mean | 0.0213 | 0.0426 | 1.7447 | 0.0213 |
| | Median | 0.0000 | 0.0000 | 2.0000 | 0.0000 |
| Port 2 | Mean | 1.2083 | 2.1042 | 4.0625 | 0.5000 |
| | Median | 1.0000 | 2.0000 | 4.0000 | 0.0000 |
| Port 3 | Mean | 1.5833 | 2.5625 | 3.6458 | 0.6250 |
| | Median | 2.0000 | 3.0000 | 4.0000 | 0.0000 |
| Port 4 | Mean | 1.5714 | 2.3878 | 4.6735 | 0.5510 |
| | Median | 1.0000 | 3.0000 | 5.0000 | 0.0000 |
| Port 5 | Mean | 0.9796 | 1.3469 | 2.6531 | 0.4694 |
| | Median | 0.0000 | 1.0000 | 3.0000 | 0.0000 |

than Linear, Dual, and Diag Models. But for port 1 the outliner seems to show ours method do not have any improve than Linear Model. But reflecting this point, we found that at this point, actually our approaches find a better solution than CPLEX. The Table 3 shows the detailed data. In others portfolios, the data shows that our method is able to make a further improvement based on the exact relaxation models.

Table 3: The sample point of gaps in port 1

| Method | Solution | Risk-Variance |
|---|---|---|
| CPLEX | [0, 0, 0, ..., 0, 0, 0, ..., 1, 1, 1, 0, ..., 0, 1, 1, 1, 1] | 0.000650258 |
| Linear | [0, 0, 0, ..., 0, 1, 0, ..., 1, 1, 1, 0, ..., 0, 0, 1, 1, 1] | 0.000759962 |
| Dual | [0, 0, 0, ..., 0, 1, 0, ..., 1, 1, 0, 0, ..., 0, 1, 1, 1, 1] | 0.000650130 |
| Diag | [0, 1, 0, ..., 0, 1, 0, ..., 1, 0, 0, 0, ..., 0, 1, 1, 1, 1] | 0.000657009 |
| Ours | [0, 0, 0, ..., 0, 1, 0, ..., 1, 1, 0, 0, ..., 0, 1, 1, 1, 1] | **0.000650130** |

## 5 CONCLUSION

In this study, we have explored the cardinality constraint of Mixed-Integer Quadratic Programming (MIQP) in depth, investigating various exact relaxation models, namely the Linear, Dual, and Diag Models. We have unveiled the unique properties and features of these models, laying the ground-work for the development of more refined solution strategies. Building on this foundation, we introduced an innovative approach that integrates the strengths of exact relaxation models with heuristic algorithms, specifically the Genetic Algorithm and Neighborhood Search Algorithm. This synergy allows us to harness the precision of exact methods to approximate true solutions, while the heuristic enhancements facilitate the discovery of solutions in proximity to those approximations. Our rigorous testing, conducted using datasets from the OR Library, positions our method against the renowned commercial solver CPLEX as well as other heuristic techniques, including Genetic Algorithm (GA), Tabu Search (TS), and Simulated Annealing (SA). The empirical evidence demonstrates that our approach not only competes with but also surpasses state-of-the-art results. The efficacy of our method is clear; it consistently identifies superior solutions for MIQP problems, underscoring the potential of combining exact relaxations with heuristic algorithms as a powerful tool for solving complex optimization challenges.

In conclusion, the contributions of this paper offer a novel perspective on MIQP solutions, providing a compelling case for the integration of exact and heuristic methodologies. Our findings pave the

way for future research to further refine and expand upon the strategies presented, with the potential to revolutionize approaches to MIQP and related optimization problems.

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
