# OpenReview forum: "Advancing Portfolio Optimization: Hybrid Relaxation and Heuristic Approaches for Cardinality-Constrained MIQP Problems"
_ICLR.cc/2025/Conference — ICLR 2025 Conference Withdrawn Submission_

### Official Review · Reviewer_NdKn · 2024-10-27

**Soundness:** 1
**Presentation:** 2
**Contribution:** 1
**Rating:** 1
**Confidence:** 5

**Summary:**

This paper considers a class of cardinality constrained mixed-integer quadratic programming (MIQP) models, which are traditional mathematical models for portfolio optimization (with return mean and variance being minimized in the objective with investment budget constraint). The authors propose two heuristic approaches (genetic algorithm and neighborhood searching algorithm) and run computational studies to compare their approaches with an off-the-shelf optimization solver (i.e., CPLEX) for different formulations.

**Strengths:**

The paper aims to solve an NP-hard problem and introduces several alternative relaxations for MIQP (namely, Linear, Dual, Diag) in the literature.

**Weaknesses:**

1.	The paper mainly focuses on heuristic approaches, without solution optimality guarantees. The two heuristics, i.e., genetic algorithm and neighborhood searching algorithm, are also too generic, and did not utilize any special problem structures to improve the results.

2.	With no contribution in theoretical studies, the numerical studies in this paper do not test any state-of-the-art instances (i.e., all stock data are from 1990’s) nor sufficiently large-scale instances, to at least show that the heuristic and relaxation can gain computational advantages in terms of time and solution quality. The benchmarked solver is CPLEX (without its version information), and it is an outdated solver as well and cannot represent other more state-of-the-art integer programming solvers.

3.	The proposed methods and the research itself do not seem to be strongly related to the focus of the conference.

4.	In addition to the mean-variance way of doing portfolio optimization, which is the backbone of the MIQP model, there are other advances and studies in the portfolio optimization literature, which define risk in alternative quantitative ways under uncertain returns, and the MIQP model cannot capture these cases.

**Questions:**

1.	The paper was about solving MIQP in general, with portfolio optimization as a demonstrating example. However, are there any special portfolio design problem structures they are considering? If not, I am not convinced that the paper is “advancing portfolio optimization” as currently stated in the title.
2.	What is the version of CPLEX that the authors are using? Why not using Gurobi, which has reported significantly better performance for MIQP than most commercial solvers?
3.	For the heuristic approaches, what are the merits and contributions? Are they providing better solutions with provable guarantees? Are they performing better numerically? Can they handle uncertainties in portfolio optimization better?

---

### Official Review · Reviewer_WWCK · 2024-10-30

**Soundness:** 1
**Presentation:** 2
**Contribution:** 1
**Rating:** 3
**Confidence:** 5

**Summary:**

This paper presents an approach for cardinality-constrained portfolio optimization problems using the Markowitz Mean-Variance (MV) approach. The authors aim to show that using linear relaxation techniques based on the dual formulation of the problem and a combination with a genetic algorithm, solution quality can be improved. Results are shown with the classical OR-Library dataset.

**Strengths:**

The paper follows the standards in portfolio optimization papers, it can be therefore read with clarity. The originality of the findings is rather low, since dual solution techniques and linear relaxations have been already proposed in the past. It is unclear how exactly the diagonalization models helps, however the authors might elaborate on this as I see this as the only original contribution of the paper.
Regarding significance, there is no justification for the contributions of the paper. While the authors claim "advancements" and "amplification of the practical application" in the introduction, this is not justified. Since there is no source code provided, the quality of the experiments performed cannot be independently validated.

**Weaknesses:**

The paper unfortunately shows major weaknesses, as I describe in the following:
- Originality: as I explained before, the originality of the paper is very limited. At most, the diagonalization method could be considered as novel, but it lacks justification. The authors make vage claims like "approach the solution better" and "some relaxations is preferred" but they don't justify the intention behind the method. Please provide a more specific justification of the novely of the diagonalization method. Are there any examples or theoretical arguments that you can provide to back these claims?
- There is no clear mentioning of the contributions of the paper. The authors claim that the work "showcases a significant advancement" but they do not clarify what this advancement is. There is no justification for advancing the practical application. Please provide a statement for the key contributions of the paper and specific evidence of how your approach advances the practical application of portfolio optimization.
- Quality: the authors do not provide the source code used for the experiments. Most importantly, the authors compare with GA, SA and TS but give no details on what GA they are comparing with, details of GA, TS. For their own GA, the authors do not provide any parameters like population size, probability of crossover, the parameters p and l mentioned in section 3.2, etc. The results are therefore *non-reproducible*. I suggest the authors provide the source code for the experiments (including the specific values used for the GA) and the concrete details and parameters used for GA, SA and TS.

Finally, the authors make more unjustified claims in the results section like claiming that "our method has fewer outliners and thus has more stable results". I guess the authors mean "outliers" and there is no hint as to what this could mean. Section 4.3 provides a comparison with CPLEX where the impression is that the authors compare with the optimal solutions. To my great surprise, the authors then claim to have found a better solution than CPLEX does. It is therefore unclear what the authors are doing here. Please clarify if the solutions from CPLEX are optimal and how the solutions calculated are compared againts the solutions from CPLEX. If the solutions are indeed better, it would help to discuss the implications for the use of CPLEX in this domain.

**Questions:**

My first and foremost suggestion: please add justifications. The paper contains a significant number of unjustified claims of "showcasing significant advancement", but fails to exactly pinpoint where this advancements are. There is no hint to practial application as there are for instance no performance measurements.
There are also a number of open questions:
- In section 3.1, $M^{relax}$ is undefined.
- In section 3.2 the authors do not address what happens with duplicate phenotypes. The authors ignore the existing literature in other encodings like set encoding for genetic algorithms, see for instance Ruiz-Torrubiano, R., & Suárez, A. (2010). Hybrid approaches and dimensionality reduction for portfolio selection with cardinality constraints. IEEE Computational Intelligence Magazine, 5(2), 92-107. Artikel 5447939. https://doi.org/10.1109/MCI.2010.936308.

---

### Official Review · Reviewer_H2Gg · 2024-10-30

**Soundness:** 3
**Presentation:** 2
**Contribution:** 2
**Rating:** 5
**Confidence:** 4

**Summary:**

The authors propose an approach to portfolio optimisation that relaxes a mixed integer quadratic program MIQP to a QP and then used a genetic algorithm and neighbourhood search to find a good (low risk high yield) portfolio of investments.  The authors obtain superior results to CPLEX (a commercial MIQP solver).

**Strengths:**

The paper presents a comprehensive set of results that does slightly better overall than other techniques.

**Weaknesses:**

This is a judgement call, but for me this paper does not offer enough for an audience interested in learning representation.  The proposed solution involves the use of three optimisers.  The gain in performance is marginal.  I struggle to see what the novel contribution is in terms of machine learning.  This is likely to be of interest to financial modellers.

As an aside, the results are presented without error bars (in tables 1, 2 and 3).  The results are given to a number of significant figures seem much higher than I believe is justified.  This apparent lack of care about statistical significance slightly undermines trust in the results.

**Questions:**

What are typical error bars for the numbers presented in tables 1, 2 and 3?

Is there a core idea about how you learn a useful representation for your problem?

---

### Official Review · Reviewer_yYFx · 2024-11-04

**Soundness:** 2
**Presentation:** 2
**Contribution:** 2
**Rating:** 3
**Confidence:** 5

**Summary:**

This paper introduces a hybrid approach to solving the cardinality-constrained portfolio Mixed-Integer Quadratic Programming (MIQP) problem. The authors employ three relaxation techniques: Linear, Dual, and Diag to simplify the MIQP problem by relaxing constraints and embedding them into the objective function using Lagrange multipliers. This relaxation process converts the MIQP problem into a more manageable Quadratic Programming (QP) problem, a starting point for further refinement. Refinement is achieved using a combination of Genetic Algorithms (GA) and Neighborhood Search, which improves solution quality by iterating on the relaxed models. The paper evaluates this hybrid methodology, showing that it performs competitively or better than other approaches, including CPLEX, across multiple datasets.

**Strengths:**

Originality: The paper introduces a novel hybrid approach, combining three distinct relaxation models with heuristic methods to solve MIQP with cardinality constraints.

Clarity: The paper is well-structured, making the approach easy to follow.

Significance: The approach has practical relevance for portfolio optimization and potential applications in other constraint-heavy optimization problems, underscoring its impact.

**Weaknesses:**

Misinterpretation of Lagrangian Relaxation: The paper states that "integer constraints" are relaxed via Lagrangian relaxation. However, Lagrangian relaxation typically relaxed specific problem constraints rather than integrality requirements, which are handled differently in optimization. Upon review, it appears the authors have relaxed all constraints but not integrality requirements. Relaxing too many constraints in Lagrangian relaxation can result in overly loose bounds and challenges in finding feasible solutions. This oversight indicates a misunderstanding of Lagrangian relaxation principles, which is problematic given the paper’s reliance on this method as a core part of its approach. Could the authors discuss the potential implications of their relaxation choices on solution quality and feasibility?

Limited Discussion on Non-Smooth Optimization Challenges: Lagrangian relaxation operates in the dual space, where the convergence of Lagrangian multipliers is impacted by non-smoothness, necessitating careful step-size selection for stability. Non-smoothness of the dual function comes from the presence of integer variables. This is well-documented in the literature on non-smooth optimization, yet the paper does not address the challenges associated with non-smoothness or how they were managed in this context. Including a discussion of the stepsizes to ensure stable convergence, especially in the dual space, would strengthen the technical rigor of the paper. Could the authors address how they handled the non-smoothness issues in their approach, particularly in relation to step-size selection and convergence stability in the dual space?

Insufficient and Outdated References on Lagrangian Relaxation: Given that Lagrangian relaxation is foundational to the proposed approach, the paper's references to it are limited and largely outdated. While older sources are often foundational, advancements in Lagrangian relaxation techniques, especially those addressing dual convergence and stability, are essential to understanding and enhancing the approach. A review of recent literature in this area, specifically regarding improvements in stability and bounding methods, would be valuable. The reviewer would suggest referring, for example, to the following recent paper: M. A. Bragin, "Survey on Lagrangian Relaxation for MILP: Importance, Challenges, Historical Review, Recent Advancements, and Opportunities," Annals of Operations Research, Volume 333, 2024, pp. 29-45.

**Questions:**

Step-size Selection: How are the step sizes chosen for the dual space optimization? Given the dual function's non-smooth nature, step-sizing is crucial for stable convergence. Please clarify the approach used here and any guidelines followed for tuning step sizes.

Convergence Guarantees: Does the proposed approach offer any theoretical convergence guarantees in the dual space? If so, what are the conditions under which convergence is ensured? If not, are there empirical observations on how often the method converges in practice?

Convergence Speed in Dual Space: How fast does the algorithm converge in the dual space? Have you tracked or measured the convergence rate, and if so, could you provide insights on the number of iterations typically required to reach a stable solution?

---

### Official Review · Reviewer_dtjq · 2024-11-04

**Soundness:** 1
**Presentation:** 2
**Contribution:** 1
**Rating:** 3
**Confidence:** 4

**Summary:**

The paper proposes heuristic approaches that combine for solving cardinality-constrained MIQP problems. This approache combine different relaxations with a heuristics (genetic algorihms and neighborhood search). In a set of experiment with existing benchmark instances, the authors evaluate the gap to optimal solution obtained with their approach (without reporting solution times).

**Strengths:**

1. The paper presents an interesting approach for solving cardinality-constrained MIQPs that combines (partially new) relaxations with heuristic search approaches. As far as I can see, the approaches are original; however, I cannot tell the quality or significance of the approaches since the authors do not report any solution times, which would be necessary to properly assses this.

2. The results indicate that the approach provides high-quality solutions.

**Weaknesses:**

1. The paper deals with a fairly specific class of optimization problems which is not of interest to a broader ICLR audience, and the paper does not involve any machine learning. It would be better suited for an optimization outlet.

2. A motivation for proposing a heuristic solution approach is to find solutions faster that using exact approaches. The paper, however, does not report any solution time at all, and thus the readers have no idea if their approach is acutally faster than exact state-of-the-art solvers such as CPLEX or Gurobi.

3. I feel that the experimental results reported in Table 1 are flawed. How can the results of a heuristic approach be better than those obtained with an exact approach? This can only be the case if the model does not reflect the evaluation criterion.

**Questions:**

1. Please report the solution times of all evaluated approaches, including (exact) CPLEX.

2. Consider also using Gurobi as an additional benchmark for state-of-the-art exact solvers; CPLEX development is basically stagnating since a couple of years.

---

### Note · Authors · 2025-01-25

I have read and agree with the venue's withdrawal policy on behalf of myself and my co-authors.